# A Novel Two-Stage Heart Arrhythmia Ensemble Classifier

Mercedeh J. Rezaei [1,*], John R. Woodward [1], Julia Ramírez [2] and Patricia Munroe [2]

1   School of Electronic Engineering Furthermore, Computer Science, Queen Mary University of London, London E1 4NS, UK; j.woodward@qmul.ac.uk
2   William Harvery Research Institute, Barts, Queen Mary University of London, London EC1M 6BQ, UK; j.ramirez@qmul.ac.uk (J.R.); p.b.munroe@qmul.ac.uk (P.M.)
*   Correspondence: m.jafarkhanloorezaei@qmul.ac.uk

**Abstract:** Atrial fibrillation (AF) and ventricular arrhythmia (Arr) are among the most common and fatal cardiac arrhythmias in the world. Electrocardiogram (ECG) data, collected as part of the UK Biobank, represents an opportunity for analysis and classification of these two diseases in the UK. The main objective of our study is to investigate a two-stage model for the classification of individuals with AF and Arr in the UK Biobank dataset. The current literature addresses heart arrhythmia classification very extensively. However, the data used by most researchers lack enough instances of these common diseases. Moreover, by proposing the two-stage model and separation of normal and abnormal cases, we have improved the performance of the classifiers in detection of each specific disease. Our approach consists of two stages of classification. In the first stage, features of the ECG input are classified into two main classes: normal and abnormal. At the second stage, the features of the ECG are further categorised as abnormal and further classified into two diseases of AF and Arr. A diverse set of ECG features such as the QRS duration, PR interval and RR interval, as well as covariates such as sex, BMI, age and other factors, are used in the modelling process. For both stages, we use the XGBoost Classifier algorithm. The healthy population present in the data, has been undersampled to tackle the class imbalance present in the data. This technique has been applied and evaluated using an ECG dataset from the UKBioBank ECG taken at rest repository. The main results of our paper are as follows: The classification performance for the proposed approach has been measured using F1 score, Sensitivity (Recall) and Specificity (Precision). The results of the proposed system are 87.22%, 88.55% and 85.95%, for average F1 Score, average sensitivity and average specificity, respectively. Contribution and significance: The performance level indicates that automatic detection of AF and Arr in participants present in the UK Biobank is more precise and efficient if done in a two-stage manner. Automatic detection and classification of AF and Arr individuals this way would mean early diagnosis and prevention of more serious consequences later in their lives.

**Keywords:** heart arrhythmia; XGBoost; classification; two-stage; machine learning; UKBiobank; atrial fibrillation; ventricular arrhythmia; multi-stage; classifier





## 1. Introduction

### 1.1. Cardiovascular Disease

There are approximately 7 million people with heart problems in the UK every year, causing around 150,000 deaths [1]. Therefore, study of cardiovascular diseases and their early detection is of the most importance. The abnormalities present in heart rhythm are known as arrhythmia. An electrocardiogram (ECG) is used to detect the heart rhythm using electrodes. Any deviations from the normal rhythm can indicate a potential underlying cardiovascular problem. Investigating and diagnosing these signals early can avoid hospitalisation or even sudden death.

### 1.2. Computer-Aided Diagnosis

Computer-aided detection or computer aided diagnosis (CAD) refers to computer-based systems that help doctors to make decisions. As the field of automatic heart arrhythmia classification uses computers to aid the doctors classify arrhythmia types, it also falls under this category. Using CAD would mean complexities such as probability to misdiagnose, human error and lack of human expertise could be eliminated or at least reduced. As a result, long-term monitoring of patients using CAD systems is highly preferred. Two examples of approaches that have used CAD are Cardiac Arrhythmia Disease Classification using the LSTM deep learning approach [2], which have used the UCI dataset for their classification purposes, and the CAD scheme for Distinguishing Between Benign and Malignant Masses on Breast images Using Deep Convolutional Neural Network that uses Bayesian Optimization [3]. In both approaches, the main goal has been to use the computer as an aid to help the doctors diagnose a disease within patients. In the first one, arrhythmia is being diagnosed and in the second one, breast cancer tissue is being diagnosed.

### 1.3. Data Set

Our data come from the UKBiobank ECG at rest repository [4]. There are 52,213 healthy participants and 1916 participants with both arrhythmia types, 162 participants with ventricular arrhythmias (Arr), and 1682 with atrial fibrillation (AF). These information were gathered in the follow-up period from the same individuals throughout their time of involvement. There are 72 cases that have both Arr and AF. Five ECG features were extracted from 10 s resting ECG of individuals with AF and Arr plus eight covariates. The features extracted include the following: RR Interval, QRS Duration, Tpe (T-peak-to-T-end) Interval, QTc (Corrected QT interval), PR interval, Sex, Age, BMI, Smoke, Diab, Chol, SBP, DBP.

### 1.4. Two-Stage Concept

The ability to detect and classify these Arr and AF correctly and quickly by using classification models with high confidence, is a goal that it yet to be achieved, at least in the real world. This is because, in theory and practice, accuracies as high as 99% have been achieved, but there is uncertainty whether these will be the same if applied in practice. The proposed two-stage model is derived from the idea of the general doctor and specialist. When a person is suspicious about some condition and they visit their general doctor, called a General Practitioner(GP) here in UK, the GP does an initial assessment and either refers the patient to a specialist or will just say there is nothing to be concerned about, depending on their diagnosis. The same concept has been introduced for our model. The first classifier acts as a GP and classifies whether a specific individual is healthy or not. The second classifier, acting similar to a specialist, then decides what disease type they have.

The major contribution of this paper is to propose a two-stage model for the classification of individuals with AF and Arr using the UK Biobank dataset with improved performance and concept over already existing models. In this paper, we describe a two-stage heart arrhythmia classification, tested on the UKBioBank data using the XGBoost Classifier. At the first stage, the pre-processed features using an Outlier Detection algorithm are classified into two main groups; normal and abnormal. At the second stage, the abnormal cases are further classified into their subgroups: AF or Arr. (Those participants with both cases AF and Arr were removed from analysis and classification for this experiment.)

### 1.5. Paper Structure

In Section 2, we discuss the background literature that is similar to this work. In Section 3, we discuss the methods used in the experiment, including the data description and the pre-processing step. In Section 4, we discuss classifiers present in our experiment. In Section 5, we discuss implementation of our classifiers, the results and evaluation of our methods. In Section 6, we conclude this paper and discuss possible future work.

## 2. Literature Background

### 2.1. Available Data Sets

There are currently two publicly available datasets present for heart arrhythmia, and these are the MIT-BIH ECG dataset and the UCI Arrhythmia dataset, and both have been used by most researchers. The downsides of these two datasets are their small size, lack of enough instances of different arrhythmia types and huge class imbalance. Therefore, in our work, we have opted to use a larger dataset in order to build more accurate classifiers.

### 2.2. Related Work

Currently, there are numerous methods present for heart arrhythmia classification. Some methods use machine learning algorithms such as support vector machines, neural networks, decision tree, K Means and other deep learning methods [5–8].

For example, Oster et al. [9] have proposed some machine learning models based on the support vector machine algorithm as well as a combination of a classical machine learning with deep learning approaches for the classification of patients with atrial fibrillation (AF). They used a subset of the UK Biobank ECG dataset for their study that is manually annotated by healthcare experts. Their combined classical machine learning and deep learning model achieved an F1 score of 84.8% on a test set and a Cohen's kappa coefficient of 0.83.

Chen et al. performed automatic heart arrhythmia classification using a combined network of Convolutional Neural Networks (CNN) and Long Short Term Memory (LSTM) using the MIT-BIH Arrhythmia dataset. Their methods demonstrated 99.32% accuracy [10].

Ashfaq Kha and Kim proposed a deep learning technique for heart arrhythmia classification using the UCI arrhythmia dataset. The approach included a noise removal method using principal component analysis (PCA) and then used LSTM for classification. They reached a classification accuracy of 93.5% using their model. This is an example of a computer-aided diagnosis as mentioned earlier the paper [2]. Fujita and Cimr proposed a CAD system for the detection of AF and flutters using a deep CNN on the MIT-BIH arrhythmia dataset. They demonstrated 98.45% for accuracy, 99.87% for sensitivity and 99.27% for specificity [11]. In a study carried out by Nishio et al., they performed lung cancer classification using deep CNN (DCNN) and transfer learning. The best averaged validation accuracies for the DCNN with transfer learning were 60.7%, 64.7% and 68.0% [12]. Al-Antari et al. used deep learning for the classification of breast cancer in digital X-ray mammograms [13]. The breast lesions were detected accurately and, and the system showed an F1-score of 99.28%. The system was also able to classify the breast lesions in less than 0.025. In another similar study of CAD, Komeda et al. used CNN to correctly classify digital Polyps images. The decisions by the classifier were correct 70% of the time [14].

In general, all the methods mentioned above show good performance for detecting and classifying ECG beats (and some of them other diseases) from a publicly available dataset such as the UCI and MIT-BIH, but they would not perform well in practice because of certain factors, including not able to generalise well, lack of enough instances and the size of the data. If and when validated using an independent external dataset, the aforementioned systems show overfitting [15]. This means their reliability in practice could be questioned, and therefore, it is difficult to analyse the stated accuracy and performance, especially when it comes to large scale databases. Therefore in this study, the UK Biobank is used to rectify this, and the two-stage model was introduced to further specialise each classifier for the diseases present in the dataset.

There are also numerous approaches that have used the idea of multi-stage. For example, Kutlu and Kuntalp [16] proposed a three-stage model using KNN classifiers. At the first stage, the heartbeats were classified into subgroups based on selected optimal features. At the second stage, the main groups were classified further into subgroups. Furthermore, lastly, at the third stage, the unclassified beats were labelled. The study used MIT-BIH arrhythmia as their dataset. The demonstrated performances were 85.59%, 95.46% and 99.56%, for average sensitivity, selectivity and specificity, respectively. In another

study, Manju and Nair [17] used a combination of SMOTEENN and XGBoost classifiers for classifying arrhythmia present in the UCI dataset. Their method uses SMOTE to address the highly imbalanced dataset problem and using XGBoost classifier as feature reduction. Their demonstrated accuracy is stated as 97.48%.

Other examples of related work include using Neural Networks in a multi-stage manner [18–20], deep genetic ensembles [21], the concept of boosting [22], using LSTM neural network [23] and using Cartesian Genetic Programming for creating an optimal digital circuit [24].

By taking into consideration the advantages and disadvantages of existing state-of-the-art models, this study proposes a two-stage classification model based on XGBoost classifier that takes in the pre-processed features. In the first stage, the classifier determines if the person belongs to the healthy or unhealthy group, and in the second stage, the classifier decides on the type of arrhythmia, including AF and Arr. In order to tackle the class imbalance problem present in the dataset, the XGBoost algorithm is used in both classifiers. The Isolation Forest algorithm is also used for outlier and noise removal. To evaluate the performance of the model, the method was validated on the UK Biobank ECG at rest arrhythmia dataset. The reason for using the XGBoost classifier depended on a few factors. One is its ability to deal with data that has class imbalance problem. A second is its ability to deal with missing data if any are present. The third is because of its speed and performance [25].

### 3. Proposed Two-Stage Method

In this section, we describe the ECG dataset and the methods used in the proposed two-stage ensemble classifier.

### 3.1. Data Description

Our data come from the UKBiobank ECG at rest repository [4]. There were 52,213 healthy participants, and 1916 participants with both arrhythmia types, 162 participants with ventricular arrhythmias (Arr) and 1682 with Atrial Fibrillation (AF). This information was gathered in the follow-up period from the same individuals throughout their time of involvement. There are 72 cases that have both Arr and AF. Five ECG features were extracted from the 10 s resting ECG of individuals with AF and Arr plus eight covariates. The features extracted include the following: RR Interval, QRS Duration, Tpe (T-peak-to-T-end) Interval, QTc (Corrected QT interval), PR interval, Sex, Age, BMI, Smoke, Diab, Chol, SBP, DBP.

### 3.2. Pre-Processing

In this section, the pre-processing of the features is explained in different subsections.

### 3.3. Missing Data

Table 1 shows the columns with missing data and their counts.

**Table 1.** Missing Data Details.

| Column Name | Missing Count |
| --- | --- |
| Tperest | 145 |
| QTc | 145 |
| QRS rest | 707 |
| BMI | 1 |
| Diab | 222 |
| smoke | 253 |
| DBP | 133 |
| SBP | 134 |
| PR | 1788 |

We have also tackled the missing data problem in our dataset. An Imputer Python library was used to fill the missing values for the numerical and categorical data [26].

### 3.4. Undersampling and Outlier Removal

Because of the significant class imbalance problem present in our dataset, we undersampled the healthy population and used XGBoost Classifier as the main algorithms in both stages. We also used an outlier detection method using the Isolation Forest algorithm. Isolation Forest is an unsupervised machine learning approach using decision trees. First, it locates the outliers by randomly selecting one of features and then carries out an arbitrary split selection between the minimum and maximum values of the feature that is selected. This process of random feature subdivision results in smaller paths being produced in the tree structure for the anomalies and therefore distinguishing them from the normal data [27]. This algorithm is relatively fast at detecting anomalies, and it requires less memory compared to other outlier detection methods; hence, it is the method we have used in our study. By adjusting the parameters of the Isolation Forest algorithm, we have removed nearly half of our healthy population. Thus, we are left with more balanced classes between healthy and unhealthy samples, reducing the imbalance by nearly 4%. The class imbalance now stands at 7% of the total population.

## 4. Classifier Description

In this section, the proposed model is explained and a flow chart of the classifier is shown in Figure 1.

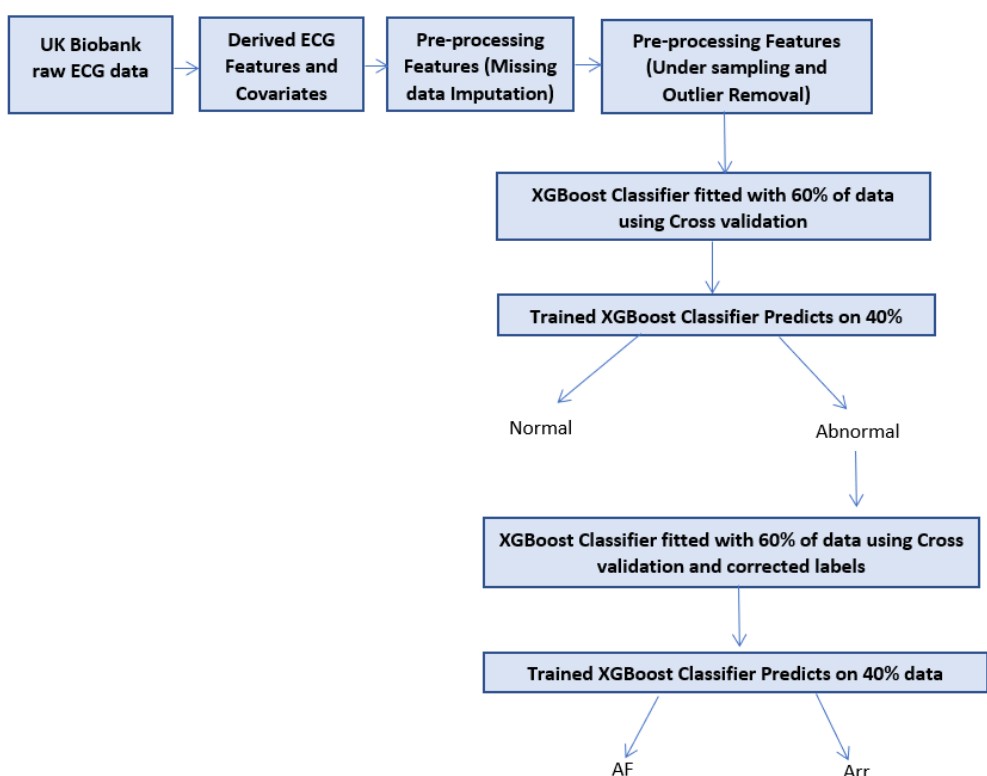

**Figure 1.** Data pre-processing and classifier training and testing Diagram.

### 4.1. Two-Stage Classifier

The proposed novel method in this paper is inspired by the diagnostic check system by doctors in the UK, where patients visit a General Practitioner, and then, if the General Practitioner indicates the patient has arrhythmia, they then visit a Specialist who tells them which form(s) of arrhythmia they suffer from. The main idea is to perform the classification in two steps using two classifiers. In the first classifier, which is the initial step of the process, the classifier separates classified cases into normal and abnormal. This is very similar to visiting a General Practitioner who does a general check and tells the patient whether there is any abnormality present and what their diagnostic is. If they

believe there is abnormality present, they would refer the patient to a specialist; this is what sparked the idea of the two-stage classifier. In the second stage, the second classifier then determines what type of disease the patient has. The classifier could also determine if they are actually healthy, and that the first classifier was wrong. This simple but powerful idea has had a magnificent impact on the classification of arrhythmia subcategories in our dataset, and we have reason to believe it will have a great impact on other problems in the healthcare industry.

### 4.2. XGBoost Classifier

There is a huge class imbalance in our data, similar to many other datasets in the healthcare industry. Only 4% of patients have heart arrhythmia, and the rest are healthy. XGBoost has been rated as one of the best machine learning algorithms in competitions for its extraordinary speed and performance and its ability to deal with missing data, skewed class distributions and a large dataset, which is exactly the data type we have. Therefore, out of all the algorithms present, we have chosen the XGBoost ensemble method [28].

XGBoost stands for Extreme Gradient boosting, and it comes from the machine learning tree models. XGBoost is an ensemble of decision tree algorithm where a pruning strategy is applied to fix the errors made by earlier made trees. Trees are added to the model until no further improvement can be seen [28,29].

### 4.3. XGBoost Parameters

In this section, for both classifiers. the XGBoost parameters are shown in Table 2. The parameters have been tuned using the python library of randomised search cross validation (cv). This method is great for finding the best combination of parameters that result in the best performance of a model, which is in essence a method of hyper parameter optimisation or hyper parameter tuning [30].

**Table 2.** XGBoost Parameters.

| Parameter | Classifier 1 Value | Classifier 2 Value |
|---|---|---|
| Objective | binary logistic | binary logistic |
| Subsample | 0.75 | 0.75 |
| No estimators | 50 | 100 |
| Colsample by tree | 0.7 | 0.7 |
| Learning rate | 0.49 | 0.99 |
| Max depth | 10 | 20 |
| Gamma | 5 | 5 |
| Alpha | 1 | 1 |
| Seed | 134 | 134 |

### 4.4. Cross Validation and Overfitting

To validate that our model is not overfitting to the test data, we have used stratified k-fold cross validation (cv) in our experiment [31]. This has been carried out for both of the classifiers. This method creates equal mean response values in all the folds by separating a different section of the train/test split each time. We used 10 splits for our k-fold cv.

## 5. Results and Discussion

### 5.1. Evaluation Metric Measures

Both of the classifiers developed in this study have been implemented using Python3 on an Intel(R) Core(TM) i7-8550U CPU 1.80GHz Laptop in the JupyterLab environment. The two-stage procedure is carried out using data from local storage. Since we used the Isolation Forest to remove outliers from our healthy population and reduce the class imbalance problem, we were left with overall data size of 27,942, each having 13 features. Therefore, total 27,942 rows were used, of which 15,942 were used for training and 12,000 used for test set. The two-stage classifier was created, and at the first stage, the system

classified the ECG feature input vectors into two main groups, normal and abnormal cases. At the second stage, the abnormal group was further classified into AF or Arr. For each classifier, the performance was measured using the F1 score, sensitivity and specificity, which are the standard statistical measures. These results are acquired by drawing a confusion matrix for each classifier and then taking the average of the results. These measures are defined as:

$$Sensitivity = TN/TN + FP \qquad (1)$$

$$Specificity = TP/TP + FN \qquad (2)$$

$$F1Score = (2 \times Sens \times Spec)/(Sens + Spec) \qquad (3)$$

*5.2. Classifiers Implementation*

In this section, the implementation of both classifiers are discussed in detail. The overall classifier has two stages: in the first stage, it separates beats into two classes, normal or abnormal. In the second stage, the abnormal group is further separated into AF or AV. In the first classifier stage, an XGBoost classifier is used. This classifier separates the two main groups from each other. In case of any contradictions between the predicted and the true label, the label is corrected before feeding it to the second stage. In the second classification stage, separated abnormal cases are classified into AF or AV. At the end of the first and second stages, the final decision is evaluated.

*5.3. Comparison with Other Methods*

The comparison of our method to other state-of-the-art models is problematic because of the difference in classification methods, data used, evaluation metrics and the type of arrhythmia's being classified. As a result, we compare the performance between this two-stage classifier with a single-stage classifier. This way, we can clearly determine if a two-stage approach is significantly different from a single-stage approach. The sensitivity and specificity of the first stage classifier are 0.785 and 0.81, respectively. The sensitivity and specificity of the second stage classifier are 0.986 and 0.909, respectively. The classifier, having separated the normal and abnormal, performs very well in detecting the arrhythmia type. If we had used the one-stage classifier without proposing this novel method, the classification would have been different. As all the classifications would have been done using one multi-class classifier, the sensitivity for AF is 0.39 with precision of 0.72 and sensitivity for Arr is 0.02 with specificity of 0.12 and for normal the sensitivity is 0.99 with specificity of 0.96. The one-stage classifier is incapable of classifying the arrhythmia types correctly. However, as mentioned earlier, because of the nature of the data, the way it has been used and the methods that apply to it, a fair comparison between the proposed two-stage classifier and other literature background is difficult [32–40].

Our results are shown in the Tables 3 and 4 below, comparing One-Stage and Two-Stage classifiers.

**Table 3.** One-Stage Classifier Results.

| Label | Sensitivity | Specificity | F1Score |
|--------|-------------|-------------|---------|
| Normal | 0.990 | 0.960 | 0.975 |
| AF | 0.390 | 0.720 | 0.506 |
| Arr | 0.020 | 0.120 | 0.034 |

**Table 4.** Two-Stage Classifier Results

| Label | Sensitivity | Specificity | F1Score |
|--------|-------------|-------------|---------|
| First-Stage | 0.785 | 0.810 | 0.797 |
| Second-Stage | 0.986 | 0.909 | 0.946 |

Table 5 shows the comparison between this model and other state-of-the-art models.

**Table 5.** Comparison with other methods.

| Method | Dataset | Sensitivity | Specificity | Accuracy | F1Score |
|:---:|:---:|:---:|:---:|:---:|:---:|
| KNN Classifier [16] | MIT-BIH | 85.59 | 95.46 | - | - |
| XGBoost and SMOTENN [17] | UCI | - | - | 97.48 | - |
| Deep learning and SVM [9] | UK Biobank | - | - | - | 84.8 |
| Random Forest [41] | MIT-BIH | - | - | 97.98 | - |
| Gradient Boosted Trees [41] | MIT-BIH | - | - | 96.75 | - |
| Our Method | UK Biobank | 98.6 | 90.9 | 99 | 94.6 |

## 6. Conclusions

This study aims to construct a heart arrhythmia classification model in a two-stage manner. The correct detection of arrhythmias present in patients results in the right medication and treatment being offered, which is vital to the patient's health improvement. The two-stage classifier has been introduced using the set of five ECG features and eight covariates, to correctly classify two types of arrhythmia, atrial fibrillation and ventricular arrhythmia. XGBoost ensemble method was used as the main detection and classification algorithm in this study. In the majority of other similar studies in the heart arrhythmia classification field, the two datasets of MIT-BIH and UCI Arrhythmia were used. These datasets are not large, and even though they include many of the arrhythmias present in the modern world; they do not contain enough examples of them. For example, the UCI arrhythmia dataset only has instances from 452 individuals. Even though it covers about 16 different arrhythmia types, as an example, there are only five examples of Atrial Fibrillation or Flutter present in the dataset [42]. In Table 6, shown below, the class code, the name of the class and the number of instances for the UCI arrhythmia dataset can be seen.

Therefore, training a classifier on such few examples means the model would not generalise well when used in practice. This is one of the main reasons we are interested in the UKBioBank ECG at rest dataset. It covers two of the most common and dangerous types of arrhythmias (AV and AF), and it has more than enough examples to be able to construct a classification system that generalises well. The ECG features and the covariates have all contributed to creating this robust method of detecting different arrhythmia types [43]. The data used and the method proposed showed satisfactory performance of classifying arrhythmia types. As the features also include background information about the patient, such as whether they are diabetic or not, smoker or not, their BMI and their blood pressure levels, this study is rated to be one of the rare studies to simulate the real environment of patients. Cardiologists, when they are looking at a patient's ECG, always take into account the underlying conditions of the patient. For example, if the patient is suffering from high blood pressure, a slight peak in their R wave would not be significantly important compared to a person who does not have any underlying conditions. The success of this two-stage classifier is also owed to the way it has been designed. The first classifier acts as a General Practitioner and just defines whether it believes the features presented to it belong to a healthy or not healthy individual. Then, in the second stage, those who have been classified as not healthy/abnormal are classified further into their correct arrhythmia type. In order to create a correct classification of the first stage, those who have been misclassified in the first stage are corrected for the second stage. This means that, for example, if the system mistakenly defined an individual as having a disease and that person is actually healthy, the label is set to its correct classification before feeding it to the second stage. This in real life could be the intervention of a doctor to distinguish between the algorithm's decision and their decision and deciding on the final verdict. Using XGBoost as the main algorithm for both stages has resulted in reduced classification time and better performance in the evaluation metrics. Another main reason for choosing the XGBoost classifier is its ability to deal with class imbalance. Even though, we removed so many outliers from the healthy population, there still a skewed class distribution present in our dataset: the class

distribution is 7% diseases and 93% healthy, and the XGBoost ensemble method has tackled this very well.

**Table 6.** UCI Arrhythmia Dataset [42].

| Class Code | Class | No of Instances |
|---|---|---|
| 01 | Normal | 245 |
| 02 | Ischemic changes (Coronary Artery Disease) | 44 |
| 03 | Old Anterior Myocardial Infarction | 15 |
| 04 | Old Inferior Myocardial Infarction | 15 |
| 05 | Sinus tachycardy | 13 |
| 06 | Sinus bradycardy | 25 |
| 07 | Ventricular Premature Contraction (PVC) | 3 |
| 08 | Superventricular Premature Contraction) | 2 |
| 09 | Left bundle branch block | 9 |
| 10 | Right bundle branch block | 50 |
| 11 | 1. degree Atrioventricular block | 0 |
| 12 | 2. degree AV block | 0 |
| 13 | 3. degree AV block | 0 |
| 14 | Left ventricle hypertrophy | 4 |
| 15 | Atrial Fibrillation or Flutter | 5 |
| 16 | Other | 22 |

In conclusion, the proposed method can be easily put into practice using similar medical data present in other fields. The classification happens in two stages. In the first stage, the first XGBoost classifier is trained on a proportion of the data using a label called "Disease" that has a binary value of 0 or 1. Those who are classified as 1 are considered to be abnormal. Then, in the second stage, these label 1s are used to classify them further into two main groups of AF and Arr. The Isolation Forest algorithm used for the detection and removal of outliers, the ensemble technique used in this two-stage method, showed that the proposed model discriminates between the two different arrhythmia types very well. The first classifier has a sensitivity of 0.785 and a specificity of 0.8. As for the second classifier, the sensitivity is 0.986 with a specificity of 0.909. The performance of the classifier is reliant on the availability of data and features present in the data. The combined classification accuracies for the proposed approach are 87.22%, 88.55% and 85.95%, for average F1 Score, average sensitivity and average specificity, respectively.

**Author Contributions:** Conceptualization, M.J.R., J.R.W. and P.M.; Data curation, M.J.R., J.R.W., J.R. and P.M.; Formal analysis, M.J.R. and J.R.W.; Funding acquisition, J.R.W.; Investigation, M.J.R.; Methodology, M.J.R.; Project administration, J.R.W.; Resources, P.M.; Software, M.J.R.; Supervision, J.R.W., J.R. and P.M.; Validation, J.R.W. and J.R.; Visualization, M.J.R.; Writing—original draft, J.R.W.; Writing—review and editing, M.J.R., J.R.W., J.R. and P.M. All authors have read and agreed to the published version of the manuscript.

**Funding:** This research was funded by the Queen Mary University of London.

**Institutional Review Board Statement:** The study was conducted according to the guidelines and approved by the Institutional Review Board of Queen Mary University Of London.

**Informed Consent Statement:** Informed consent was obtained from all subjects involved in the study.

**Data Availability Statement:** The data used for this experiment come from the UK BioBank ECG at rest repository. These data are not available to the public. An application has to be made to be able to use or download the data.

**Acknowledgments:** I would like to acknowledge that the data used for this paper come from the UKBiobank with the application number of 8256. I would also like to express my thanks and gratitude to other members of the Electrogenomics group who helped with the ECG markers derivation, with special thanks to Dr. Stefan van Duijvenboden and Michele Orini. Julia Ramírez also

acknowledges support from the European Union's Horizon 2020 research and innovation program under the Marie Sklodowska-Curie grant agreement no 786833.

**Conflicts of Interest:** None declared.

**Abbreviations**

The following abbreviations are used in this manuscript:

| | |
|---|---|
| Arr | Ventricular Arrhythmia |
| AF | Atrial Fibrillation |
| TP | True Positives |
| TN | True Negatives |
| FP | False Positives |
| FN | False Negatives |

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
