# Peer review of "A Novel Two-Stage Heart Arrhythmia Ensemble Classifier"

_computers, doi:10.3390/computers10050060_

Round 1
Reviewer 1 Report
Dear authors,
This paper is about a two stages classifier for heart arrhythmia. It uses the XGBoost Classifier algorithm in the first step for classifying the samples in normal or abnormal, and in a second step for classifying the abnormal samples in atrial Fibrillation or Ventricular 5 Arrhythmia (Arr).
The dataset used was the UKBioBank ECG.
This is an interesting topic and it may be really useful in a real context.
However, there are some issues that must be improved before publication:
- The introduction does not provide many information
- The background has several issues:
- Line 29: “..paper. ome..”
- Line 31: confusing sentence
- Line 32: cannot understand if reference [2] is for the sentence before or after the period. The references should appear before the period at the end of a sentence.
- Line 32: Your say "in all the papers", but only present one reference at the end of the sentence
- Line 38: and etc.[6] [7] [8] [9][10] [11] - Etc means nothing to me. Please describe all the studies.
- Line 40: add a reference for XGBoost Classifier
- Line 43: Please define AF and Arr in the text
- Line 45: You are already at the end of section 2. If you want to describe the outline of your paper, please do it at the end of the introduction, not at the end of the related work.
- There is a lot of information in the conclusion that should appear in the related work.
- Materials and Methods issues:
- Line 56: Is reference 12 a website? Please clarify this.
- Line 64: please define all the abbreviations
- Line 66: “we we” typo
- Classifier Description
- Line 89: “the classifier separates classifies cases” typo
- Results and Discussion
- Can you guarantee that there isn’t overfitting for the samples in the test set?
- Conclusion
- Novel information should not be included in the conclusion.
- A table called a figure
Author Response
Hi there,
Thanks for your comments. Please see the attachment.

Reviewer 2 Report
In this paper, the authors proposed a two-stage ensemble classifier model for the classification of heart arrhythmia. At the first stage, features of the ECG input classified into two main classes; normal and abnormal. At the second stage, these abnormal cases are further classified into two specific diseases. The following comments are listed for the authors to further improve the paper quality.
- The Introduction section is too short. More contents about the heart arrhythmia, classification of heart arrhythmia could be added.
- In this paper, only two specific diseases, Atrial Fibrillation (AF) and Ventricular Arrhythmia (Arr), are used in the classification. As there are more other heart related diseases, an explanation is needed in the paper.
Author Response

(The authors gave the same response as above.)

Reviewer 3 Report
This paper describes a novel two-stage ensemble classifier model for the classification of heart arrhythmia. The method consists of two stages. At the first stage, features of the ECG input classified into two main classes: normal and abnormal. At the second stage, these abnormal cases are further classified into two specific diseases: Atrial Fibrillation (AF) and Ventricular Arrhythmia (Arr). A diverse set of ECG features, as well as covariates, are used throughout the modeling process. For both stages, the classifiers are based on the XGBoost Classifier algorithm. This technique has been applied to an ECG dataset from the UKBioBank ECG taken at the rest repository. The classification accuracy for the proposed approach has been measured using the F1 score, Sensitivity (Recall), and Specificity (Precision). the paper is interesting overall, but the following are the comments that must be addressed:
Comments:
- Authors need to re-write the Abstract in a more meaningful way example (Problem definition=> How existing methods are lacking => proposed solution => Outcome).
- Authors should elaborate technically on why for both stages, the classifiers are based on the XGBoost Classifier algorithm is selected instead of several other Machine learning classifiers.
- The introduction should be revised, the current version has too short this part, as paper related to computer aided diagnosis so authors should add few lines of CAD and should include these references.
Ahmadi, A., Kashefi, M., Shahrokhi, H., & Nazari, M. A. (2021). Computer-aided diagnosis system using deep convolutional neural networks for ADHD subtypes. Biomedical Signal Processing and Control, 63, 102227. M. A. Khan and Y. Kim, "Cardiac arrhythmia disease classification using lstm deep learning approach," Computers, Materials & Continua, vol. 67, no.1, pp. 427–443, 2021.
Hizukuri, Akiyoshi, Ryohei Nakayama, Mayumi Nara, Megumi Suzuki, and Kiyoshi Namba. "Computer-Aided Diagnosis Scheme for Distinguishing Between Benign and Malignant Masses on Breast DCE-MRI Images Using Deep Convolutional Neural Network with Bayesian Optimization." Journal of Digital Imaging 34, no. 1 (2021): 116-123.
- The major contribution of the paper is missing.
- The authors should draw the block diagram of the proposed approach that depicts step by step analysis of the proposed approach.
- Related work section 2 must be updated. At least 45 references and more than 60% references should be from the last three years.
The authors include most of the references out of date. Please update it. Better to include Table which elaborate what is difference between your approach and previous researchers.
- AUTHORS should need to give all experiments parameters still few parameters are missing??
- 3. Comparison with other methods, authors should draw Table Here.
- UKBiobank ECG Data set has an imbalance issue how authors tackle this issue?
- The structure of the paper is very poor and there are a lot of typo mistakes authors should think about it deeply.
Author Response
- Authors need to re-write the Abstract in a more meaningful way example (Problem definition=> How existing methods are lacking => proposed solution => Outcome).
REPLY: Thanks for the above comment. It has greatly enhanced the paper. The entire abstract has been rewritten to address the comments and to make it more meaningful.
- Authors should elaborate technically on why for both stages, the classifiers are based on the XGBoost Classifier algorithm is selected instead of several other Machine learning classifiers.
REPLY: Thanks for the above comment, we greatly appreciate the feedback we have received from you. The reason behind choosing the XGBoost classifier is explained in the literature background section the last paragraph. We had also tested other algorithms such as SVM, random forest before making the decision to use XGBoost as the main algorithm for both classifiers. We had also tried different combinations of algorithm.
- The introduction should be revised, the current version has too short this part, as paper related to computer aided diagnosis so authors should add few lines of CAD and should include these references.
REPLY: Thank you for the above feedback. We have rewritten the introduction and literature background sections and have added the necessary references (including the ones shown below in your original comment) for CAD systems in place. We want to thank you deeply as the above comment has enhanced the quality of our paper.
Ahmadi, A., Kashefi, M., Shahrokhi, H., & Nazari, M. A. (2021). Computer-aided diagnosis system using deep convolutional neural networks for ADHD subtypes. Biomedical Signal Processing and Control, 63, 102227. M. A. Khan and Y. Kim, "Cardiac arrhythmia disease classification using lstm deep learning approach," Computers, Materials & Continua, vol. 67, no.1, pp. 427–443, 2021.
Hizukuri, Akiyoshi, Ryohei Nakayama, Mayumi Nara, Megumi Suzuki, and Kiyoshi Namba. "Computer-Aided Diagnosis Scheme for Distinguishing Between Benign and Malignant Masses on Breast DCE-MRI Images Using Deep Convolutional Neural Network with Bayesian Optimization." Journal of Digital Imaging 34, no. 1 (2021): 116-123.
- The major contribution of the paper is missing.
REPLY: Thanks for the above feedback, we have added the major contribution and the aims of this paper on line 74 – 82.
- The authors should draw the block diagram of the proposed approach that depicts step by step analysis of the proposed approach.
REPLY: Thanks for the above constructive comment, the block diagram has been added on page 6; Figure 1. We have tried to be as through as possible with the diagram and show all steps one by one of the proposed approach. We believe the above comment has greatly enhanced the paper quality so that readers can comprehend the model in more detail.
- Related work section 2 must be updated. At least 45 references and more than 60% references should be from the last three years.
REPLY: Thanks for the above constructive comment, we have rewritten the entire literature background and now have around 45 references from the most recent papers. We really appreciate getting this feedback as it has greatly changed the structure and concept of the paper in a positive way.
The authors include most of the references out of date. Please update it. Better to include Table which elaborate what is difference between your approach and previous researchers.
REPLY: Thanks for the above constructive comment. We have updated the references that were out of date. The table of comparison with other approaches has also been added on page 9, Table 5.
- AUTHORS should need to give all experiments parameters still few parameters are missing??
REPLY: Thanks for the above constructive comment. We have given the XGBoost parameters for both classifiers in table 2 on page 7. We believe that by adding this parameters in place, we have been able to help with reader’s comprehension of the paper and model.
- 3. Comparison with other methods, authors should draw Table Here.
REPLY: Thanks for the above constructive comment. We have addressed this in table 6 on page 9.
- UKBiobank ECG Data set has an imbalance issue how authors tackle this issue?
REPLY: Thanks for the above constructive comment. We have addressed this in section 3.4.
- The structure of the paper is very poor and there are a lot of typo mistakes authors should think about it deeply.
REPLY: Thanks for the above constructive comment. In order to improve the paper structure, we have made so many changes. We have broken down the introduction to different subsections with more information. We have also changed the literature background greatly. We have also changed other sections on the paper again by giving more details, separating paragraphs and adding more meaningful subsections to appropriate sections to improve the paper flow and structure.
Round 2
Reviewer 3 Report
The authors did excellent work and resolve all my previous comments now this paper looks very good for the readers so I agree to accept this paper for publication in the present form.